# Gradual Emergence of *East African cassava mosaic Cameroon virus* in Cassava Farms in Côte d'Ivoire

**Bekanvié S. M. Kouakou** [1,2,*], **Aya Ange Naté Yoboué** [1,2], **Justin S. Pita** [1,2], **J. Musembi Mutuku** [2], **Daniel H. Otron** [1,2], **Nazaire K. Kouassi** [1,2], **Kan Modeste Kouassi** [1,2], **Linda Patricia L. Vanié-Léabo** [1,2], **Cyrielle Ndougonna** [2], **Michel Zouzou** [1] and **Fatogoma Sorho** [1]

1     UFR Biosciences, Université Félix Houphouët-Boigny (UFHB), Abidjan 22 BP 582, Côte d'Ivoire; angenate60@gmail.com (A.A.N.Y.); pita.wave.ci@gmail.com (J.S.P.); danyotron452@gmail.com (D.H.O.); kouassinazaire@gmail.com (N.K.K.); kanmodestekouassi@gmail.com (K.M.K.); lindaleabo@hotmail.fr (L.P.L.V.-L.); zouzoum2001@yahoo.fr (M.Z.); fsorho@gmail.com (F.S.)

2     The Central and West African Virus Epidemiology (WAVE) for Food Security Program, Pôle Scientifique et d'Innovation, Université Félix Houphouët-Boigny (UFHB), Abidjan 22 BP 582, Côte d'Ivoire; josiahmutuku@gmail.com (J.M.M.); cyrielle.ndougonna@wave-center.org (C.N.)

\*     Correspondence: bekanviemarie@yahoo.fr

**Abstract:** Cassava mosaic disease (CMD) and cassava brown streak disease (CBSD) are endemic threats to cassava production, causing significant yield losses. They are caused respectively by begomoviruses and ipomoviruses that are transmitted by whiteflies and infected cuttings. This study aimed to monitor and characterize viruses responsible for these diseases in order to fill existing gaps in understanding their epidemiology in Côte d'Ivoire. Field surveys were conducted in 2016, 2017, and 2020, and no CBSD symptoms were observed. However, an increase in CMD incidence was noted from 45.95% in 2016 to 51.37% in 2020, while CMD severity remained moderate over the years with a mean value of 2.29. The number of whiteflies was relatively low and decreased over the years. Molecular diagnostics carried out on cassava leaf samples allowed the detection of *East African cassava mosaic Cameroon virus* (EACMCMV) that occurs in single as well as in mixed infection with *African cassava mosaic virus* (ACMV). Single EACMCMV infection, which was detected only in three agroecological zones of eastern Côte d'Ivoire in 2016, spread throughout the country in 2017 and became more widespread in 2020 with a preponderance in central and southern zones, whereas ACMV and EACMCMV coinfection has spread to the entire zones. Phylogenetic analysis of the viral isolates showed that they are closely related to those from Burkina Faso, Ghana, and Nigeria. This changing population of cassava virus species constitutes a serious threat to cassava cultivation. Understanding the evolutionary dynamics of these viruses could help in adopting better disease management strategies to control the disease.

**Keywords:** cassava; epidemiology; CMD; begomoviruses; Côte d'Ivoire





## 1. Introduction

Cassava (*Manihot esculenta* Crantz) is an important starchy root crop grown globally in tropical and subtropical regions [1]. It is now considered a potential solution to the impending food crisis in Africa because it offers the greatest resilience to climate change [2]. Africa contributes approximately 64.7% of the world's cassava production of 314.8 million tons/year [3]. In West Africa, cassava production is estimated at 96.2 million tons/year and contributes to 33% of African production [3]. Côte d'Ivoire is the third-highest cassava-producing country in West Africa after Nigeria and Ghana [3], and cassava is the second-most-consumed food crop in Côte d'Ivoire after yams. With a production of over 6.5 million tons of fresh cassava tubers in 2020, according to the Food and Agriculture Organization of the United Nations [3], cassava is of immense economic importance in Côte d'Ivoire.

Cassava plays a significant role in human nutrition as a source of food (such as attiéké, tapioca, cookies, pasta, gari, etc.) and serves as a crucial ingredient in various industrial products (including starch, biofuel glues, glucose, etc.) as well as serving as a feed base for livestock. The production of this important root crop is seriously threatened by two viral diseases, cassava mosaic disease (CMD) and cassava brown streak disease (CBSD), which are considered the major disease constraints in sub-Saharan Africa [4]. While CMD is widespread across Africa, CBSD is found in Eastern and Central Africa [5]. CMD is caused by begomoviruses with yield losses between 50% and 70% [6], whereas CBSD is caused by ipomoviruses and results in total crop loss of up to 100% in susceptible cultivars [7,8]. These viruses are transmitted either by whitefly vectors (*Bemisia tabaci*) or by the use of diseased planting materials [9]. CMD symptoms are observed only on the leaves of infected plants, causing patchy leaf chlorosis with little or no mottling in cases of mild infection and severe chlorosis, smaller leaves, and stunting when the infection is severe [10]. CBSD symptoms, on the other hand, occur on all parts of the plant (leaves, stem, and tuberous roots). These diseases can spread very rapidly and escalate to serious pandemics. Thus, in the 1990s, an epidemic of unusually severe CMD emerged in Uganda and subsequently spread to several countries and large areas in East and Central Africa [11]. A novel recombinant begomovirus, East African cassava mosaic virus–Uganda (EACMV-UG), was shown to be associated with this epidemic [12,13]. Almost 30 years after the first reports of severe CMD from Uganda, the implicated EACMV-UG continues to spread, currently advancing southward through the eastern Democratic Republic of Congo and westward through central Cameroon [14]. According to the International Committee on Taxonomy of Viruses (ICTV), eleven cassava mosaic begomoviruses have been described, of which nine occur in Africa, either alone or in combination, i.e., African cassava mosaic virus (ACMV), African cassava mosaic Burkina Faso virus (ACMBFV), East African cassava mosaic virus (EACMV), East African cassava mosaic Cameroon virus (EACMCMV), East African cassava mosaic Kenya virus (EACMKV), East African cassava mosaic Malawi virus (EACMMV), East African cassava mosaic Zanzibar virus (EACMZV), cassava mosaic Madagascar virus (CMMGV), and South African cassava mosaic virus (SACMV). The two other viruses that occur in Asia are Indian cassava mosaic virus (ICMV) and Sri Lankan cassava mosaic virus (SLCMV) [15,16]. Dual or multiple members of the cassava begomovirus group can participate in mixed infections, typically characterized by severe symptoms [17]. Nationwide surveys conducted to date in West African countries have shown that three of the nine virus species described in Africa (ACMV, EACMV, and EACMCMV) are the viruses most commonly found in cassava fields [13,18–22]. In Côte d'Ivoire, EACMCMV was first reported in 2001 [13], and it was always found to be associated with ACMV. The presence of these viruses was also documented later in cassava in Côte d'Ivoire by [18,19]. Despite these studies, knowledge of the epidemiology of cassava mosaic viruses in Côte d'Ivoire is still poor. Additionally, the most devastating cassava viral disease, cassava brown streak disease (CBSD), which has been moving westward over the years, has already been reported in the Democratic Republic of Congo (DRC) [23]. In addition, the recombinant virus from Uganda, EACMV-UG, is spreading to neighboring countries [24] and has been detected in DRC, eastern Gabon [25], and Burkina Faso [26], which is near Côte d'Ivoire. If these pathogens manage to cross the barriers of Côte d'Ivoire, where cassava is of vital importance to the population's nutrition, they could decimate the fields and cause famine, as has been the case in Uganda. To prevent this scourge and to be well equipped in case it happens, it is important to monitor these diseases.

This study was carried out to fill the gaps in quality scientific data and scientific evidence necessary for policy-driven anticipation, preparedness, and rapid response against cassava viral outbreaks and epidemics in Côte d'Ivoire.

## 2. Materials and Methods

### 2.1. Study Area

Surveys were conducted in 2016, 2017, and 2020 across six of the seven agroecological zones of Côte d'Ivoire (Figure 1) [27], located between the latitudes 4°30′ and 10°30′ N. The agroecological zones I, II, IV, and V are characterized by two dry seasons (from July to August and December to March) and two rainy seasons (from September to November and April to July). Agroecological zone III is characterized by a short dry season (from November to December) and a long rainy season (from March to October), while agroecological zone VI is characterized by a long dry and a short rainy season, from November to April and from May to October, respectively. The seventh zone was not surveyed because of the very rare presence of cassava fields in this area.

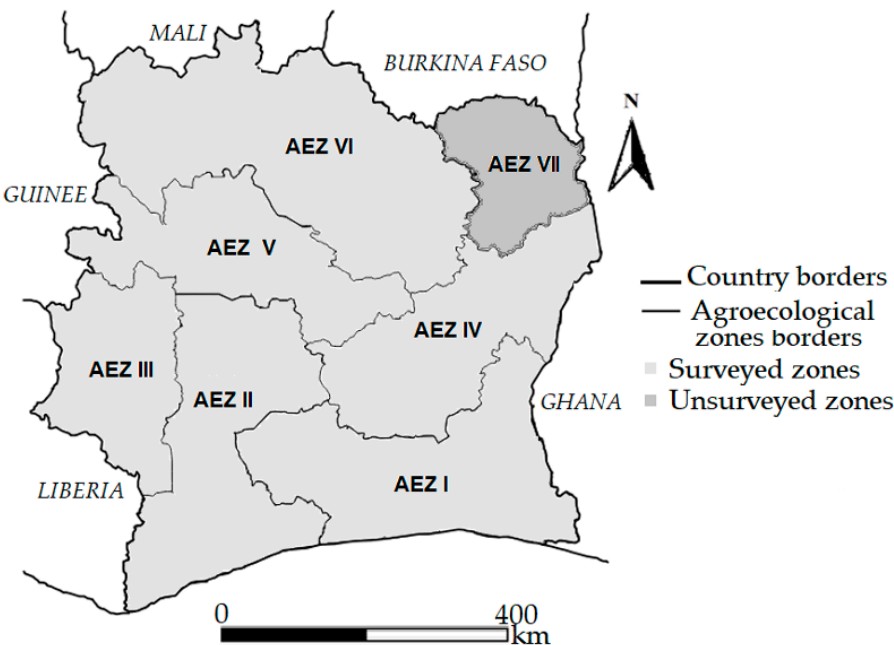

**Figure 1.** Locations of the six agroecological zones surveyed.

### 2.2. CMD Epidemiological Assessment

2.2.1. Survey Data Collection

Surveys were conducted in 854 localities throughout the six agroecological zones surveyed following the Central and West African Virus Epidemiology Program (WAVE) harmonized protocol [28]. Cassava fields, aged from 3 to 6 months, were surveyed approximately every 10 km along the main roads. In each site surveyed, information such as the age of the field, the source of the planting material, and the names of cassava varieties were recorded. A Cassava Survey App (iForm) co-developed by WAVE and the University of Cambridge (UK) was used to collect epidemiological data. In each field, 30 cassava plants were assessed randomly along two diagonals to form an "X" pattern ($15 \times 2 = 30$ plants per field). Individual plants were assessed visually for the presence or absence of CMD symptoms, the severity of the symptoms was scored, the number of whiteflies on the top five leaves of each plant was counted, and the mode of CMD infection was assessed. For each field, the CMD incidence mean, the severity mean, and the mean whitefly population were calculated.

Cassava leaf samples were collected according to the different symptom severity levels present in each field assessed (no symptoms, mild, severe, and very severe symptoms). On the other hand, young leaves were given priority when collecting samples from symptomatic plants, as viral rates were higher than in old leaves. The various samples were kept in herbarium form at room temperature [22,28].

The CMD incidence (I) per field was determined by calculating the ratio of diseased plants to the total number of plants assessed according to the following formula [28]:

$$I (\%) = 100 \times (\text{Number of diseased plants}/\text{Total number of plants observed}) \quad (1)$$

CMD incidence in the field was then classified based on in-field visual assessment as healthy (0%), low incidence (0–25%), medium incidence (25–50%), high incidence (50–75%), and very high incidence (75–100%).

For each cassava plant, CMD symptom severity was assessed based on a published scale in which disease severity levels were scored from 1 (no symptoms) to 5 (most severe symptoms) based on visual assessment [20,29–31], as shown in Figure 2. Mean CMD severity per field was then calculated according to the formula below [22,28]:

$$Sm = \Sigma \ (\text{number of diseased plants} \times \text{corresponding infection score})/\text{Total number of diseased plants among the 30 assessed} \quad (2)$$

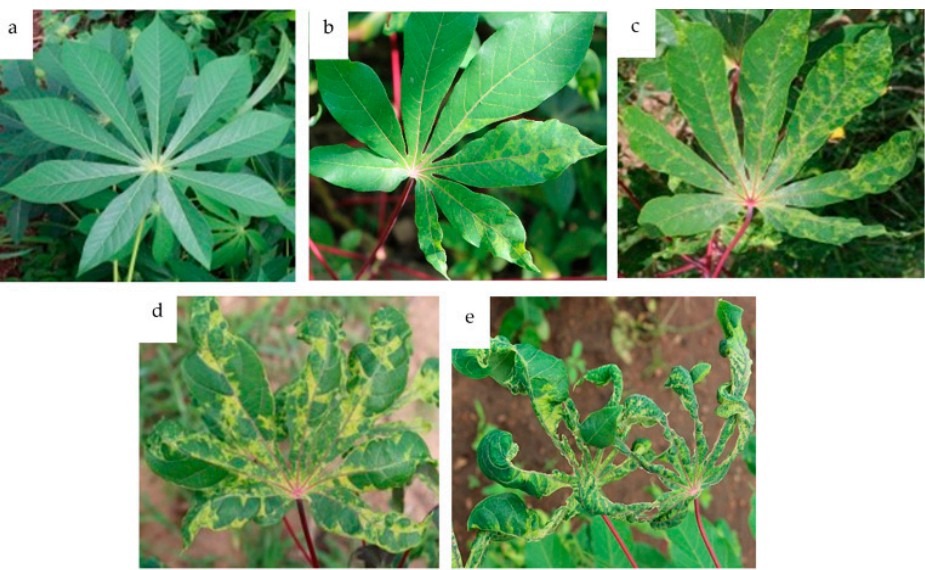

**Figure 2.** Cassava leaves showing different stages of symptom severity in the field, using a scale from 1 (no symptoms) to 5 (very severe symptoms): (**a**) = score 1, (**b**) = score 2, (**c**) = score 3, (**d**) = score 4, and (**e**) = score 5.

### 2.2.2. Statistical Analysis

Epidemiological data were recorded in iForm software v.6.9.3. and then uploaded into the WAVE Cube, holding centrally all the field survey data collected in the different countries of the WAVE network. These data were then imported into R software (Version 3.5.1, R Development Core Team, Vienna, Austria, 2010). The effect of agroecological zones was tested on CMD incidence, severity, and whitefly abundance using a generalized linear model (*glm*) with likelihood-ratio test (chi-squared test or Fisher test in case of overdispersion) and Tukey's pairwise mean comparison test ($\alpha = 0.05$).

### 2.3. PCR Diagnostic of Cassava Mosaic Begomoviruses

Total DNA was extracted from cassava leaves using the CTAB protocol previously described by Doyle and Doyle [32]. The DNA concentration of each sample was determined using a spectrophotometer (Eppendorf) and adjusted to 50 ng/μL for PCR. The DNA amplification was performed by PCR using virus-specific primer pairs for detection of ACMV, EACMV, and EACMCMV (Table 1). The PCR was conducted in a total volume of 25 μL containing 1 × Reaction Buffer, 0.5 mM of dNTP (NEB), 0.2 mM MgCl$_2$, 0.5 μM each primer (Eurogentec, Seraing, Belgium), 0.625 U of Taq polymerase, and 5 μL of DNA template (about 150 ng). Amplification conditions included a first step of denaturation at

94 °C for 5 min. This initial denaturation step was followed by 35 cycles of 1 min at 94 °C, 1 min at 52 °C, and 3 min at 72 °C, and then a final elongation step at 72 °C for 10 min. The PCR products were separated on a 1% agarose gel electrophoresis, stained in ethidium bromide, and viewed under a UV gel imager.

**Table 1.** Primers used for ACMV, EACMV, and EACMCMV detection.

| Primers Names | Sequences (5′-3′) | Target Region | Size | Reference |
|---|---|---|---|---|
| JSP 001<br>JSP 002 | ATGTCGAAGCGACCAGGAGAT<br>TGTTTATTAATTGCCAATACT | ACMV DNA-A (CP) [1] | 783 bp | [17] |
| ACMVBF<br>ACMVBR | TCGGGAGTGATACATGCGAAGGC<br>GGCTACACCAGCTACCTGAAGCT | ACMV DNA-B (BV1/BC1) | 628 bp | [33] |
| JSP 001<br>JSP 003 | ATGTCGAAGCGACCAGGAGAT<br>CCTTTATTAATTTGTCACTGC | EACMV DNA-A (CP) | 780 bp | [17] |
| CMBRepF<br>EACMVRepR | CRT CAA TGA CGT TGT ACC A<br>GGT TTG CAG AGA ACT ACA TC | EACMV DNA-A (AC1) | 650 bp | [34] |
| VNF031F<br>VNF032R | GGATACAGATAGGGTTCCCAC<br>GACGAGGACAAGAATTCCAAT | EACMV-CM DNA-A (AC2/AC3) | ≈560 bp | [35] |

[1] CP = capsid protein.

### 2.4. Sequencing and Phylogenetic Analysis

PCR products were sequenced by GENEWIZ (Germany). Contigs were assembled and edited using Geneious Prime® 2022.2.1. (Biomatters Ltd, Auckland, New Zealand.) software. Consensus sequences obtained were subjected to BLASTn Search in NCBI for the virus identity. The representative sequences of the various cassava begomoviruses were downloaded from the Genbank for phylogenetic analyses. Sequence alignments were performed with the MUSCLE and CLUSTAL W algorithms in MEGA X software [36]. The phylogenetic tree was generated using the maximum-likelihood (ML) methods with the general time-reversible (GTR) model as the best-fit model for substitution pattern description. The robustness of individual branches was tested by bootstrap analysis [37] performed using 1000 replicates.

## 3. Results

### 3.1. CMD Incidence in 2016, 2017, and 2020

During the study period, characteristic symptoms of CMD were observed in almost all cassava fields assessed, with only 3.28% (28/854) of the surveyed fields showing no signs of mosaic disease. CMD symptoms were observed in 96% (154/160), 97% (334/344), and 96.6% (338/350) of the fields assessed in 2016, 2017, and 2020, respectively. Out of the fields surveyed across Côte d'Ivoire during the four-year span of this study, 18.5% (158/854) had a low CMD incidence (0–25%), 30.21% (258/854) had a medium CMD incidence (25–50%), 26.58% (227/854) had a high CMD incidence (50–75%), and 21.43% (183/854) had a very high CMD incidence (75–100%). The mean incidence of CMD was $45.95 \pm 0.27\%$ (SE) in 2016, $50.32 \pm 0.28\%$ in 2017, and $51.37 \pm 0.25\%$ (SE) in 2020 (Figure 3a). CMD incidence varied greatly between the six agroecological zones, and this difference was highly significant each year ($p = 1.16 \times 10^{-7}$). The highest average incidence of CMD was recorded in agroecological zones I and III, in southern and western Côte d'Ivoire, in agroecological zone VI in northern Côte d'Ivoire and in agroecological zone II in southwestern in 2016, 2017, and 2020, respectively. It is worth noting that the lowest incidence was reported in agroecological zone IV in the far eastern part of the country for all three years (Figure 3b).

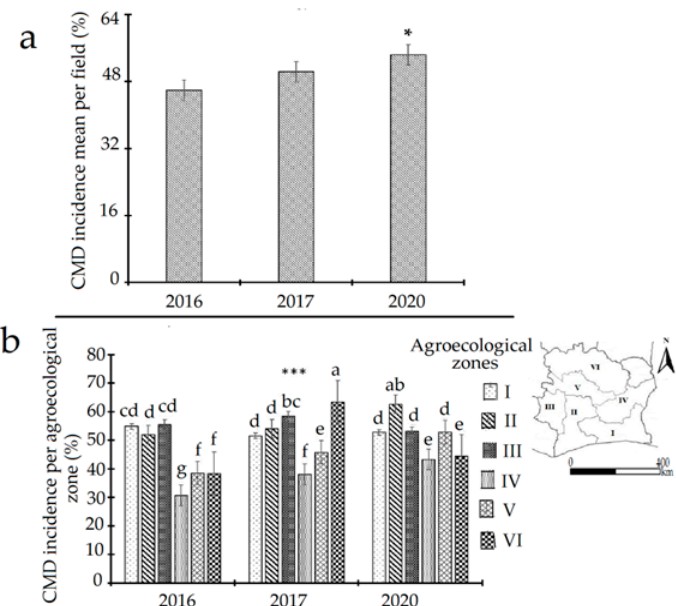

**Figure 3.** Cassava mosaic disease incidence according to the surveyed years (**a**) and agroecological zones (**b**) in Côte d'Ivoire. Data are means ± SE. *** represents a very highly significant difference at $p \leq 0.001$ and * represents significant difference at $p \leq 0.05$. The bars represent the standard error. Bars sharing the same letters are not significantly different between years and between agroecological zones (generalized linear model (*glm*) and Tukey's pairwise mean comparison test $p < 0.05$).

*3.2. CMD Severity and Mode of Infection*

The severity of CMD symptoms observed in the field was generally moderate, with an average severity score of 3 over the four years. However, significant differences were noted between the agroecological zones with a *p*-value of 0.000607. Indeed, during the 2016 surveys, agroecological zone I recorded the highest mean CMD severity score (2.61), and zone VI recorded the lowest (2.01). This trend was reversed in 2020, with zone VI recording the highest average severity (2.72) and zones I and IV having the lowest average severity score (2.27). In 2017, all zones recorded almost the same average severity (Figure 4).

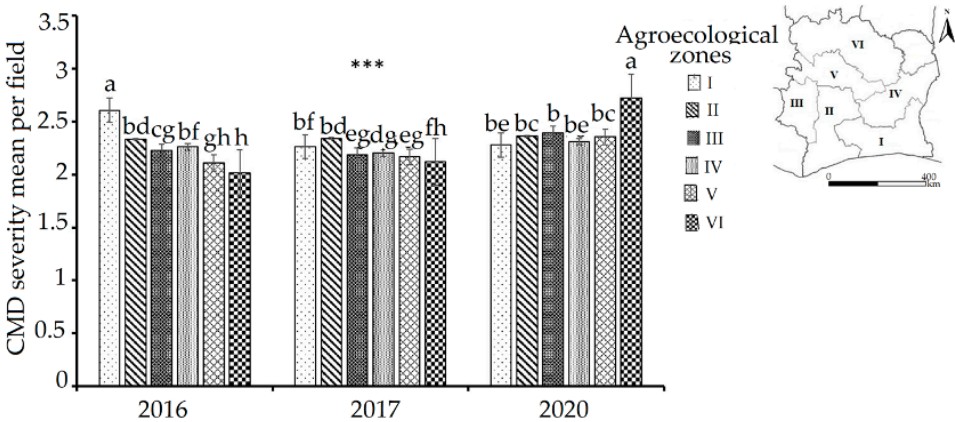

**Figure 4.** Cassava mosaic disease severity by agroecological zones during the years 2016, 2017, and 2020 in Côte d'Ivoire. Data are means ± SE. *** represents a very highly significant difference at $p \leq 0.001$. The bars represent the standard error. Bars sharing the same letters are not significantly different between years and between agroecological zones (generalized linear model (*glm*) and Tukey's pairwise mean comparison test $p < 0.05$).

Assessment of CMD infection mode observed during this study showed that cutting-borne infection was more prevalent than whitefly-borne infection during each of the three

years. Over the three years, whitefly-transmitted infections ranged from 2 to 10% of the total infections observed, while cutting-transmitted infections ranged from 90 to 98%. The rate of cutting-transmitted infection remained statistically identical ($p = 0.211$) during each of the three years, whereas the rate of whiteflies transmitted infections showed a significant difference between years with $p = 0.0151$.

### 3.3. Abundance of Whiteflies in Cassava Fields Surveyed

There was a very significant difference between mean whitefly counts per plant in 2016, 2017, and 2020 ($p = 2 \times 10^{-16}$). These differences were also very significant between agroecological zones ($p = 2.43 \times 10^{-12}$). Whitefly mean decreased from about six whiteflies per plant in 2016 to less than two whiteflies per plant in 2020. The highest values were observed in agroecological zone I in 2016 and 2017 and in agroecological zones I, II, and V in 2020. Moreover, the lowest mean whitefly count per plant was recorded in agroecological zones III and V during the 2016 and 2017 surveys and only in agroecological zone III in 2020 (Figure 5).

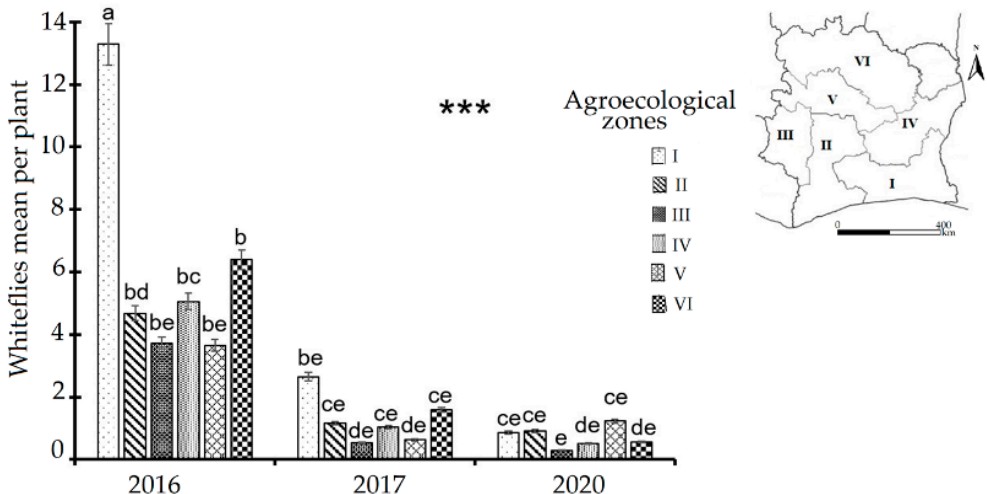

**Figure 5.** Whitefly abundance according to the year and by agroecological zone in Côte d'Ivoire. Data are means ± SE. *** represents a very highly significant difference at $p \leq 0.001$. The bars represent the standard error. Bars sharing the same letters are not significantly different between years and between agroecological zones (generalized linear model (*glm*) and Tukey's pairwise mean comparison test $p < 0.05$).

### 3.4. Evolution and Distribution of Cassava Mosaic Begomoviruses

Two cassava mosaic begomoviruses were identified in this study. Except for *African cassava mosaic virus* (ACMV), the use of the specific pair of primers VNF031/VNF032 confirmed that all the amplified EACMV in this study were *East African cassava mosaic Cameroon virus* (EACMCMV). These viruses were detected in 73.52% (322/438) of cassava leaf samples collected in 2016. Among these, 73.52% (322/438) were infected by cassava mosaic begomoviruses. Out of these, 35.8% were infected by ACMV alone, 5.9% were infected by EACMCMV alone, and 31.7% were coinfected by ACMV and EACMCMV. This infection rate decreased in 2017 (61.8%; 498/806). Of this, around 33.9% of samples were infected by ACMV alone, 3.6% of samples were infected by EACMCMV alone, and around 24.3% were coinfected with ACMV and EACMCMV. A similar trend was observed in 2020, with an infection rate decreased again to 59.2% (500/844). This rate was segregated as follows: 16.9% of samples were infected by ACMV alone, 9.2% were infected by EACMCMV alone, and 33.1% were coinfected by both ACMV and EACMCMV (Table 2).

**Table 2.** PCR results obtained from samples collected during 2016, 2017, and 2020 surveys in Côte d'Ivoire.

| Survey Years | Viruses Detected | | | | |
|---|---|---|---|---|---|
| | Number of Samples Tested | ACMV | EACMCMV | ACMV/EACMCMV | Negative |
| 2016 | 438 (100%) | 157 (35.84%) | 26 (5.94%) | 139 (31.74%) | 116 (26.48%) |
| 2017 | 806 (100%) | 273 (33.87%) | 29 (3.60%) | 196 (24.32%) | 308 (38.21%) |
| 2020 | 844 (100%) | 143 (16.94%) | 78 (9.24%) | 279 (33.06%) | 344 (40.76%) |

The single ACMV infection rate decreased from 2016 (35.8%) to 2020 (16.9%). Similar trends were observed for single EACMCMV infection and coinfection with ACMV and EACMCMV between 2016 and 2017. However, between 2017 and 2020, both single EACMCMV infection and coinfection virus rates increased (Table 2).

The three-year CMB distribution maps showed different infection trends from one year to another according to their repartitioning in each agroecological zone. Our study revealed that ACMV was the dominant virus found in all regions of cassava cultivation in 2016. But between 2017 and 2020, we observed a regression in the rate of ACMV single infection, whereas EACMCMV single infection occurrence increased in all agroecological zones. Also, we noticed that ACMV single infections were gradually replaced by coinfection of ACMV and EACMCMV, indicating that EACMCMV is gaining ground over ACMV. Mixed infections of ACMV and EACMCMV were more prevalent in southern and central Côte d'Ivoire (Figure 6).

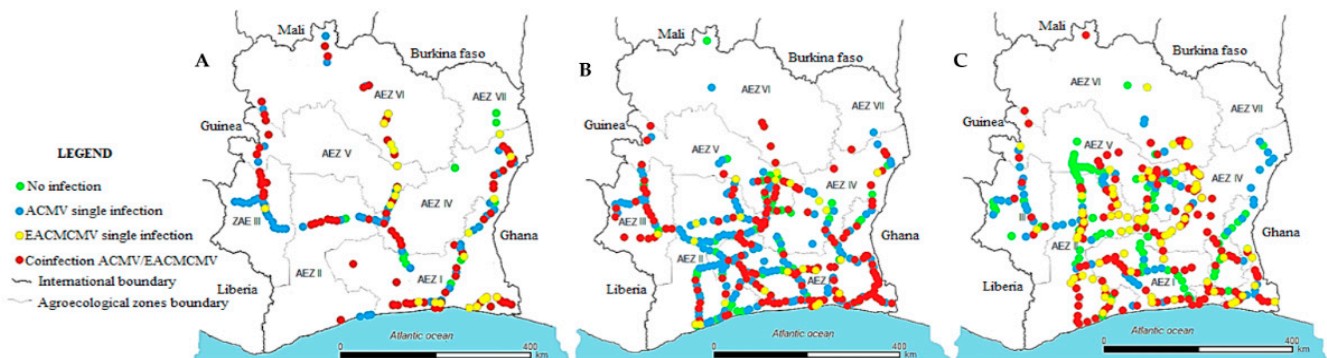

**Figure 6.** Distribution of *African cassava mosaic virus* and *East African cassava mosaic Cameroon virus* in single and mixed infections in different agroecological zones of Côte d'Ivoire: (**A**) 2016, (**B**) 2017, and (**C**) 2020.

### 3.5. Phylogenetic Analysis of ACMV and EACMCMV Coat Protein Genes

Phylogenetic analysis of CP sequences obtained from PCR products with specific primers confirmed that the viruses circulating in Côte d'Ivoire are ACMV and EACMCMV.

Indeed, ACMV isolates identified shared the highest nucleotide identity (97–99%) with ACMV isolates from Ghana (MG250164), Côte d'Ivoire (AF259894), Burkina Faso (LC659083, LC658964), and Benin (KR476371). In turn, the investigated EACMCMV isolates from this study were most closely related to the EACMCMV isolates from Ghana (MG250164, JN165089), Côte d'Ivoire (AF259896), Nigeria (EU685326), Madagascar (KJ887944), and Burkina Faso (LC659083) with nucleotide identities between 97 and 99% (Figure 7).

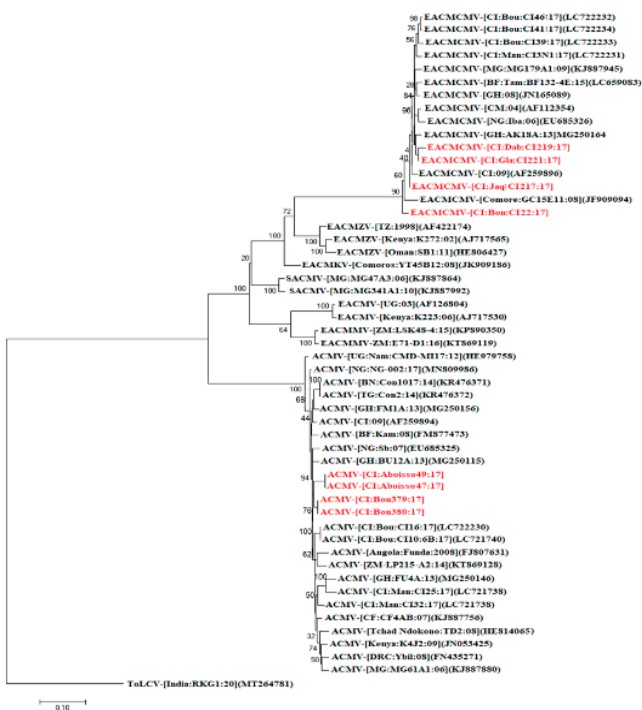

**Figure 7.** Maximum-likelihood phylogenetic tree of *African cassava mosaic virus* (ACMV) and *East African cassava mosaic Cameroon virus* (EACMCMV) based on multiple alignment of nucleotide sequences of the coat protein. Analyzed isolates are indicated by red color. As an outgroup, the sequences of TLCV were used. The accession numbers of used isolates are indicated. The scale bar represents a genetic distance.

## 4. Discussion

This study provides important information on the spread of CMD and the involved viruses in Côte d'Ivoire.

The three-year field survey data revealed that CMD is widespread across the six agroecological zones in Côte d'Ivoire, with incidence increasing from 46% to 51%. This finding indicates that farmers are not sufficiently aware of the existence of CMD, and no phytosanitary measures are taken to renew the cassava fields. This contributes to increasing the impact of the disease and threatens food security, given the importance of cassava for small-scale farmers. Chikoti et al. [6] made a similar observation in Zambia, where farmers persist in spreading the disease by utilizing infected cuttings from vulnerable local cultivars. Moreover, our study has shown a particularly high CMD incidence in the south (AEZ I, AEZ II, and AEZ III) and central part of the country (AEZ V) in contrast to the center-east (AEZ IV), where the impact of the disease was relatively low. Indeed, AEZ I and II are traditional zones of high production and high consumption of cassava. AEZ III is also an area of high cassava cultivation, while AEZ V represents the new region of intensive cassava cultivation with big farms in Côte d'Ivoire. The high CMD incidence observed in these zones could be linked to the use of local cultivars, which are more susceptible to CMD [6,38]. The high incidence of CMD in these important cassava-growing areas is a major concern, and effective measures need to be taken to avoid a shortage of cassava in Côte d'Ivoire. It should be noted that this high CMD incidence over the years was not generally associated with very severe symptoms. The average severity of the CMD was moderate in the fields. It is known that in endemic areas, CMD incidence can be high without the symptoms being very severe [39]. Our findings are similar to those of Ntawuruhunga et al. [40] in the Republic of Congo and those of Zinga et al. [41] in the Central African Republic, who also reported a very high incidence of CMD with moderate severity of symptoms.

Furthermore, our data suggest that the primary factor contributing to the spread of CMD in Côte d'Ivoire is the utilization of infected cassava cuttings. Indeed, farmers typically rely on planting materials provided by their neighbors' farms or from their own old farms regardless of their phytosanitary status and, therefore, contribute to spreading the disease. These farming practices are widespread in sub-Saharan Africa, with consequences for the spread of diseases in root and tuber crops such as manioc [24,41–45].

A relatively low abundance of whiteflies was recorded during the surveys, confirming that whiteflies are unlikely to be a key factor in the spread of CMD in Côte d'Ivoire. Additionally, we observed a decrease in the mean number of whiteflies per field over the study period. A similar observation was recently made in Burkina Faso [20]. However, AEZ I recorded a relatively higher average number of whiteflies than the other zones. Since cropping practices are not necessarily the same in the different zones, this could be the reason for these observations. Indeed, Fondong et al. [46] showed that whiteflies were more abundant in fields where cassava was grown alone than in those where it was grown in association with other crops. In AEZ I, almost all the cassava fields visited were found without any association with other crops. On another view, the preference of certain cassava varieties over others by whiteflies could explain their unequal distribution between zones. Cassava varieties cultivated in some zones were more attractive to whiteflies than others. According to Pastório et al. [47], cultivars with dark green leaves are the most attractive to whiteflies.

Molecular analysis of cassava leaf samples collected from the fields in 2016, 2017, and 2020 confirmed the presence of ACMV and EACMCMV in Côte d'Ivoire, as previously reported by Pita et al. [17] and Toualy et al. [18]. ACMV was detected countrywide and was the most prevalent virus [19]. However, the rate of occurrence of single ACMV infections dropped from 2017 to 2020 and seems to be progressively replaced by the coinfection of ACMV and EACMCMV. The high prevalence of both viruses revealed in this study could lead to a considerable reduction in the yield, as it has been shown by Owor et al. [48] that cassava plants in which mixed infections were detected had a significantly lower yield than those infected with a single virus. The synergistic interaction between ACMV and EACMV-UG was a key factor in the Ugandan epidemic in the 1990s [13]. Here also, coinfection of ACMV and EACMCMV is becoming more and more prevalent. These findings are very alarming because the situation here presents high similarities with the explosive combination of factors that triggered the devastating CMD epidemic in Uganda. The third key driver of the Ugandan epidemic, the surge of the whitefly population, is not a significant factor in Côte d'Ivoire because of the low abundance of whiteflies recorded through the four-year duration of this study. Furthermore, we are reporting that the rate of single EACMCMV infection is increasing with time and is spreading within Côte d'Ivoire. The Ugandan CMD epidemic was correlated with the occurrence of a recombinant virus, EACMV-UG, that spread 20 km southward every year from the center of the country. Likewise, we have here a recombinant virus, EACMCMV, with an increasing proportion of the cassava viral population. This recombinant was always found in mixed infection with ACMV [4,17,34,49] but is now found in an increasing proportion of single infections, suggesting it may have acquired additional function(s) enabling the virus to infect cassava plants alone. Phylogenetic analysis of the sequences using specific primer pairs indicated that ACMV and EACMCMV isolates found in this study were very closely related to species circulating in West Africa. This is probably the result of planting material exchange between Côte d'Ivoire and the neighboring countries such as Burkina Faso, Ghana, and Nigeria, where ACMV and EACMCMV have also been reported [22,24,50].

## 5. Conclusions

Considering all of these findings, farmers must receive training on how to recognize cassava diseases, how to select appropriate cuttings for establishing new plots, and on all phytosanitary measures they must adhere to prevent the spread of the disease. Moreover,

continuous monitoring of these diseases is required to avoid an outbreak of an epidemic, which could seriously compromise food security in Côte d'Ivoire.

**Author Contributions:** Conceptualization, J.S.P., N.K.K. and F.S.; methodology, J.S.P.; software, D.H.O. and B.S.M.K.; validation, J.S.P.; formal analysis, D.H.O. and B.S.M.K.; investigation, B.S.M.K., A.A.N.Y., C.N., N.K.K. and F.S.; resources, J.S.P.; data curation, B.S.M.K.; writing—original draft preparation, B.S.M.K.; writing—review and editing, J.S.P., N.K.K., J.M.M., L.P.L.V.-L., K.M.K., C.N., M.Z. and F.S.; visualization, J.S.P.; supervision, J.S.P., F.S. and N.K.K.; project administration, J.S.P.; funding acquisition, J.S.P. All authors have read and agreed to the published version of the manuscript.

**Funding:** This research was funded by the Bill and Melinda Gates Foundation and the United Kingdom Foreign, Commonwealth, and Development Office (FCDO; INV-002969; grant no. OPP1212988) to the Central and West African Virus Epidemiology (WAVE) Program for root and tuber crops, Université Félix Houphouët-Boigny (UFHB). Under the grant conditions of the foundation, a Creative Commons Attribution 4.0 Generic License has already been assigned to the author-accepted manuscript version that might arise from this submission.

**Data Availability Statement:** The raw data supporting the conclusions of this article will be made available by the authors on request.

**Acknowledgments:** The authors also want to thank the WAVE team for helping with the data collection and laboratory analysis and Fidèle Tiendrébéogo and Angela Eni for improving the manuscript.

**Conflicts of Interest:** The authors declare no conflicts of interest.

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
