# Peer review of "Gradual Emergence of East African cassava mosaic Cameroon virus in Cassava Farms in Côte d’Ivoire"

_agronomy, doi:10.3390/agronomy14030418_

Round 1

Reviewer 1 Report

Comments and Suggestions for Authors

Assuming the quality of the experimental part, figures, and the main text this paper can be accepted. However, I have one suggestion that could be incorporated to the manuscript. It is advisable to include a visual representation illustrating CMD symptoms at different severity levels for the readers' better understanding.

Author Response

Comments: Assuming the quality of the experimental part, figures, and the main text this paper can be accepted. However, I have one suggestion that could be incorporated to the manuscript. It is advisable to include a visual representation illustrating CMD symptoms at different severity levels for the readers' better understanding.

Response: Thank you for the suggestion. The manuscript was updated accordingly. Pictures showing CMD symptoms were added in the Figure 2, between lines 166 and 174.

Reviewer 2 Report

Comments and Suggestions for Authors

In the manuscript entitled “Gradual emergence of East African cassava mosaic Cameroon virus in cassava farms in Côte d’Ivoire” the Authors focused on the epidemiology of cassava mosaic disease (CMD) and cassava brown streak disease (CBSD) in Côte d'Ivoire. Their nationwide field surveys conducted in 2016, 2017, and 2020 revealed the absence of CBSD but an increase in CMD incidence. Molecular analysis indicated the presence and rapid expansion of East African cassava mosaic Cameroon virus (EACMCMV) and its co-infection/ interactions with African cassava mosaic virus (ACMV). The study emphasizes the need for enhanced disease management strategies based on a comprehensive understanding of virus dynamics across the country.

In my opinion, the undertaken studies are very important, especially because cassava is the most widely cultivated crop on the continent, and the control of investigated viruses is crucial for securing food supplies and maintaining food security. These diseases can significantly decrease cassava yields, with important consequences for millions of people who depend on this root as a staple in their diet.

In general, the manuscript is written rather fairly logically, however, there are a few significant shortcomings. Therefore, this manuscript requires major revision and improvement before publication.

Below I presented the major comments:

I suggest reorganizing the abstract to become more informative and coherent. Please point out the major aims of this study (monitoring of viruses, molecular diagnosis followed by phylogeny studies), Afterwards, briefly summarize the main results, followed by conclusions and interpretation. Please explain also the role of whiteflies as vectors in virus epidemiology, because this fragment is now inconsistent.

In the introduction section, the author effectively introduces the topic, but please emphasize why it is so important to monitor these viruses that cause such dangerous diseases. The result section is written in an understandable and detailed way. My main comment is about the discussion. I think it's written in a very chaotic and unclear manner. Please rephrase this section. Some fragments are replicated. Additionally, it would be beneficial to provide more in-depth explanations and connections between the research findings and their implications for food security in Africa.

 Minor comments

Line 33 double-space

Line 43 delete dot after )

Lines 71-72 …can participate in mixed infections, typically characterized by severe symptoms.

Line 77 …is still poor.

Lines 89-90 delete this sentence and citation after mentioned latitudes as well the referring to Fig.1

Line 121 …..protocol [25]. Please delete described by…, or previously described by and put the names of authors according to author guidelines

Line 132 symptomatic and asymptomatic

Line 133 as previously described [citation] This error appears many times. Please review the whole text and make appropriate corrections

Line 146 I don’t understand exactly this formula, what is the difference between total score and total number?

Lines 147-152 short this sentence, maybe collected data was deposited in iFroms… etc.

Line 194 retrieved - downloaded?

Line 224 double-space

Line 273-278 change line spacing

Line 202-205 change: During the study period, characteristic symptoms of CMD were observed in almost all cassava fields assessed, with only 3.28% (28/854) of the surveyed fields showing no signs of mosaic disease

Lines 310-315 Reorganized this fragment to improve it.

Figure 3,4,5 What is the significance of the small letters on the bars? Additionally, the legend regarding the markings of the studied agroecological zones is illegible.

Figure 5 please change the title, is unclear

Line 356 our- collected /identified (in this study )

Line 358 in turn, investigated in this study EACMCMV

Figure 7 changes the title Maximum likelihood phylogenetic tree of African … and East African….based on the multiple alignments of nucleotide sequences of the coat protein. Analyzed isolates are indicated by red colour. As an outgroup, the sequences of TLCV were used. The accession numbers of used isolates are indicated. The scale bar represents a genetic distance.

Author Response

Comment 1: I suggest reorganizing the abstract to become more informative and coherent. Please point out the major aims of this study (monitoring of viruses, molecular diagnosis followed by phylogeny studies), Afterwards, briefly summarize the main results, followed by conclusions and interpretation. Please explain also the role of whiteflies as vectors in virus epidemiology, because this fragment is now inconsistent.

Response: Thank you for the suggestion. The abstract has been reorganized and we think it is now more coherent and informative (Line 14 to Line 32).

“Cassava mosaic disease (CMD) and cassava brown streak disease (CBSD) are endemic threats to cassava production causing significant yield losses. They are transmitted by whiteflies and infected cuttings. This study aimed to monitor and characterize viruses responsible for these diseases, in order to fill the existing gaps in understanding their epidemiology in Côte d’Ivoire. Surveys were conducted in 2016, 2017 and 2020 and no CBSD symptoms was observed in any fields. However, an increase of CMD incidence was observed from 45.95% in 2016 to 51.37% in 2020 while CMD severity remained moderate over the years with a mean value of 2.29. Whiteflies number was relatively low and decreased over the years. Molecular diagnostic carried out on cassava leaf samples allowed detection of East African cassava mosaic Cameroon virus (EACMCMV) that occurs in single as well as in mixed infection with African cassava mosaic virus (ACMV). Single EACMCMV infection, which was detected only in three agroecological zones of the eastern Côte d’Ivoire in 2016, spread throughout the country in 2017 and became more widespread in 2020 with a preponderance in the central and southern zones, whereas the ACMV+EACMCMV coinfection has spread to the entire zones. Phylogenetic analysis of the viral isolates showed that they were closely related to those from Burkina Faso, Ghana and Nigeria. This changing population of cassava virus species constitutes a serious threat to cassava cultivation. knowledge of the evolutionary dynamics of these viruses could help to adopt better disease management strategies to control the disease.”

Comment 2: In the introduction section, the author effectively introduces the topic, but please emphasize why it is so important to monitor these viruses that cause such dangerous diseases. 

Response 2: Thank you for pointing this out. It is important to monitor these viruses to avoid those which are not yet in the country and moreover better manage those that are always there to don’t spread more (Line 85 to Line 93).

Comment 3: My main comment is about the discussion. I think it's written in a very chaotic and unclear manner. Please rephrase this section. Some fragments are replicated. Additionally, it would be beneficial to provide more in-depth explanations and connections between the research findings and their implications for food security in Africa.

Response 3: Thank you for your comment. The discussion has been reorganized (Line 432 to 508)

Comment 4: Line 33 double-space

Response 4: The additional space has been deleted (Line 38).

Comment 5: Line 43 delete dot after )

Response 5: The dot after ) has been deleted (Line 48)

Comment 6: Lines 71-72 …can participate in mixed infections, typically characterized by severe symptoms.

Response 6: These changes have been done (Line 77)

Comment 7: Line 77 …is still poor.

Response 7: This comment has been taken into account (Line 84)

Comment 8: Lines 89-90 delete this sentence and citation after mentioned latitudes as well the referring to Fig.1

Response 8: This sentence and the citation have been deleted. However, the title of the figure has been maintained. This is a very important figure that shows the location of the different agroecological zones surveyed (Line 100).

Comment 9: Line 121 …..protocol [25]. Please delete described by…, or previously described by and put the names of authors according to author guidelines

Response 9: This comment has been taken into account (Line 128).

Comment 10: Line 132 symptomatic and asymptomatic

Response 10: The sentence containing this error has been reworded (Line 139 to 140).

Comment 11: Line 133 as previously described [citation] This error appears many times. Please review the whole text and make appropriate corrections

Response 11: Thank you for pointing out it. It has been revised in the whole manuscript (Line 155).

Comment 12: Line 146 I don’t understand exactly this formula, what is the difference between total score and total number?

Response 12: The formula has been rewritten to make it easier to understand (Line 157-158). Sm = Σ (number of diseased plants X corresponding infection score) / Total number of diseased plants among the 30 assessed

Comment 13: Lines 147-152 short this sentence, maybe collected data was deposited in iFroms… etc.

Response 13: This sentence has been shortened (Line 177 to 179).

Comment 14: Line 194 retrieved - downloaded?

Response 14: This change has been made (Line 220).

Comment 15: Line 224 double-space

Response 15: The additional space has been deleted (Line 235).

Comment 16: Line 273-278 change line spacing

Response 16: The line spacing has been changed (Line 296 to 302).

Comment 17: Line 202-205 change: During the study period, characteristic symptoms of CMD were observed in almost all cassava fields assessed, with only 3.28% (28/854) of the surveyed fields showing no signs of mosaic disease

Response 17: Thank you for the improvement of this sentence. It has been taken account (Line 228 to 230).

Comment 18: Lines 310-315 Reorganized this fragment to improve it.

Response 18: This paragraph has been reorganized (340 to 356).

Comment 19: Figure 3,4,5 What is the significance of the small letters on the bars? Additionally, the legend regarding the markings of the studied agroecological zones is illegible.

Response 19: The small letters on the bars were significant at p<0.05 (Lines 266-294-336).

Comment 20: Figure 5 please change the title, is unclear

Response 20: The title of this figure has been changedWhitefly abundance according to the year and by agroecological zone in Côte d’Ivoire” (Line 333).

Comment 21: Line 356 our- collected /identified (in this study)

Response 21: This change has been made (Line 404).

Comment 22: Line 358 in turn, investigated in this study EACMCMV

Response 22: This change has been made (Line 406 to 407).

Comment 23: Figure 7 changes the title Maximum likelihood phylogenetic tree of African … and East African….based on the multiple alignments of nucleotide sequences of the coat protein. Analyzed isolates are indicated by red colour. As an outgroup, the sequences of TLCV were used. The accession numbers of used isolates are indicated. The scale bar represents a genetic distance.

Response 23: Thank you for your suggestion. The title of this figure has been changed according to what you proposed (Line 432 to 508).

Reviewer 3 Report

Comments and Suggestions for Authors

This manuscript describes a three-year survey of the occurrence and seriousness of cassava mosaic disease (CMD) in Côte d’Ivoire. Three begomoviruses, namely  African cassava mosaic virus (ACMV), East African cassava mosaic virus (EACMV) and East African cassava mosaic Cameroon virus (EACMCMV), that were believed to be associated with CMD were detected by PCR. Based on the relative trend of the occurrence of the viruses, authors conclude that “ACMV single infections were gradually replaced by mixed infection ACMV+EACMCMV indicating that EACMCMV is getting ground over ACMV”. The study is of high interest to readers in cassava production in Africa, particularly in Côte d’Ivoire, and the survey appears to be comprehensive and reasonably well conducted. However, there are many issues that need to be clarified. Following are examples of the issues (please note, they are just examples; and authors themselves need to find and fix those):
Abstract:

-          It needs to be revised to better reflect the relative significance of major components of the study;

-          Virus names should be written in accordance to the ICTV guidelines: https://ictv.global/filebrowser/download/440. This point applies to the entire manuscript;

Introduction:

-          It would be useful (and necessary as far as I am concerned) if a certain level of background information (or even justification) on why only ACMV, EACMV and EACMCMV were targeted by PCR assay while nine begomoviruses have been known to infect cassava crops in Africa;

Materials and Methods:

-          Leaf sample collection (L132-133): More details should be provided. This information is very important because it affects the PCR data interpretation (section 3.4);

Results:

-          The results shown in Fig. 2  should be incorporated into 3 by adding one bar for the average incidences in it;

-          The results described in Lines 248-251 need to be moved forward and placed right before description of the mean CMD values.

Author Response

Comment 1:  Abstract needs to be revised to better reflect the relative significance of major components of the study;

Response 1: Thank you for this comment. The abstract has been restructured to reflect the major component of the study (Line 14 to 32).

Comment 2: Virus names should be written in accordance to the ICTV guidelines: https://ictv.global/filebrowser/download/440. This point applies to the entire manuscript;

Response 2: Viruses names have been revised according ICTV guidelines (Lines 1 and 74).

Comment 3: Introduction would be useful (and necessary as far as I am concerned) if a certain level of background information (or even justification) on why only ACMV, EACMV and EACMCMV were targeted by PCR assay while nine begomoviruses have been known to infect cassava crops in Africa;

Response 3: These viruses are the most commonly found in west African countries cassava fields according to data provided from nationwide surveys conducted by more authors (Line 78 to 81).

Comment 4: Leaf sample collection (L132-133): More details should be provided. This information is very important because it affects the PCR data interpretation (section 3.4);

Response 4: Thank you for pointing this out. This section has been improved (Line 139 to 143).

Comment 5: The results shown in Fig. 2 should be incorporated into 3 by adding one bar for the average incidences in it;

Response 5: Thank you for your suggestion. These Figures have been combined (Line 244-261).

Comment 6: The results described in Lines 248-251 need to be moved forward and placed right before description of the mean CMD values.

Response 6: This change has been done (Line 230 to 235).

Round 2

Reviewer 2 Report

Comments and Suggestions for Authors

Dear Authors,

the revised manuscript that was submitted looks much better now. The Authors have addressed all the comments and clarified ambiguities, and additionally, new figures have been added, which has also improved the overall quality of the work. However, I have noted a few minor comments that I would like the Authors to consider before the publication of the text.

Lines 15-16 in the abstract: please rewrite this sentence to avoid implying that insects directly transmit the disease, as it is understood that the causative agents are viruses carried by these insects. The same comment – line 55

Line 19-20 was observed…was observed  (repeating)

Line 27 maybe: mixed infection of ACMV and EACMCMV, or coinfection of both viruses…I'm not sure about using the + sign (the same comment lines: 481, 486,)

Line 29 that they are…

Line 31 Rewrite the sentence ( suggestion: Understanding the evolutionary dynamics of these viruses could help in adopting better disease management strategies to control the disease.)

Line 33 Maybe it would be worth adding the keyword cassava

Line 47 for consideration: Cassava plays a significant role in human nutrition as a source of food (such as Attiéké, tapioca, cookies, pasta, gari, etc.) and serves as a crucial ingredient in various industrial products (including starch, biofuel, glues, glucose, etc.), as well as serving as a feed base for livestock.

Line 100 in the previous version you cited position [24] when presenting agroecological zones and I commented only on the way of this citation, maybe it is worth adding this information in the description under the figure (Figure 1. Locations of the six agroecological zones previously described by Leeg et al. ) or add citation after Figure 1 [24] (in a main text )

Line 407 in this study…

Line 439 I suggest rewriting the sentence ( my suggestion: Chikoti et al. [37] made a similar observation in Zambia, where farmers persist in spreading the disease by utilising infected cuttings from vulnerable local cultivars.)

Line 445 instead of belt I suggest using region, but it is only a suggestion…

Line 457: My suggestion: Furthermore, our data suggested that the primary factor contributing to the spread of CMD in Côte d’Ivoire is the utilization of infected cassava cuttings.

Line 458: farmers typically rely on planting materials… regardless of their phytosanitary….

Line 464: Additionally, we observed a decrease in the mean number of whiteflies per field over the study period. A similar observation was recently made in Burkina Faso [24].

Line 504 Considering all of these findings, farmers must receive training on how to recognize cassava diseases, how to select appropriate cuttings for establishing new plots, and on all phytosanitary measures they must adhere to prevent the spread of the disease.

Author Response

Thank you very much for taking your time to review this manuscript. Please find the responses below and the corresponding revisions/corrections highlighted in the re-submitted files.

Comments 1: Lines 15-16 in the abstract: please rewrite this sentence to avoid implying that insects directly transmit the disease, as it is understood that the causative agents are viruses carried by these insects. The same comment – line 55

Response 1: The sentences have been rewritten.They are caused respectively by begomoviruses and ipomoviruses that are transmitted by whiteflies and infected cuttings”. Line 15-16 and Line 57.

Comments 2: Line 19-20 was observed…was observed (repeating)

Response 2: The second “observed” has been replaced by “noted”. Line 20.

Comments 3: Line 27 maybe: mixed infection of ACMV and EACMCMV, or coinfection of both viruses…I'm not sure about using the + sign (the same comment lines: 481, 486,)

Response 3: Thank you for your suggestions. This has been taken into account throughout the document where the + sign appears between ACMV and EACMV. Lines 28, 356, 358, 374-375, 474, 479.

Comments 4: Line 29 that they are…

Response 4: This has been changed. Line 29.

Comments 5: Line 31 Rewrite the sentence (suggestion: Understanding the evolutionary dynamics of these viruses could help in adopting better disease management strategies to control the disease.)

Response 5: This suggestion has been taken into account. Line 31-33.

Comments 6: Line 33 Maybe it would be worth adding the keyword cassava

Response 6: This key word has been added. Line 34.

Comments 7: Line 47 for consideration: Cassava plays a significant role in human nutrition as a source of food (such as Attiéké, tapioca, cookies, pasta, gari, etc.) and serves as a crucial ingredient in various industrial products (including starch, biofuel, glues, glucose, etc.), as well as serving as a feedbase for livestock.

Response 7: This sentence has been improved according to your suggestion. Line 48-51.

Comments 8: Line 100 in the previous version you cited position [24] when presenting agroecological zones and I commented only on the way of this citation, maybe it is worth adding this information in the description under the figure (Figure 1. Locations of the six agroecological zones previously described by Leeg et al. ) or add citation after Figure 1 [24] (in a main text )

Response 8: This reference has been added after Figure 1. Line 102.

Comments 9: Line 407 in this study…

Response 9: This has been removed. Line 397.

Comments 10: Line 439 I suggest rewriting the sentence (my suggestion: Chikoti et al. [37] made a similar observation in Zambia, where farmers persist in spreading the disease by utilising infected cuttings from vulnerable local cultivars.)

Response 10: This sentence has been changed according to your suggestion. Line 432-434.

Comments 11: Line 445 instead of belt I suggest using region, but it is only a suggestion…

Response 11: According to your suggestion, belt has been replaced by region. Line 438.

Comments 12: Line 457: My suggestion: Furthermore, our data suggested that the primary factor contributing to the spread of CMD in Côte d’Ivoire is the utilization of infected cassava cuttings.

Response 12: Thank you for this suggestion. It has been taken into account. Line 449-450.

Comments 13: Line 458: farmers typically rely on planting materials… regardless of their phytosanitary….

Response 13: These changes have been made. Line 450-452.

Comments 14: Line 464: Additionally, we observed a decrease in the mean number of whiteflies per field over the study period. A similar observation was recently made in Burkina Faso [24].

Response 14: This sentence has been rewritten. Line 456-458.

Comments 15: Line 504 Considering all of these findings, farmers must receive training on how to recognize cassava diseases, how to select appropriate cuttings for establishing new plots, and on all phytosanitary measures they must adhere to prevent the spread of the disease.

Response 15: Thank you for this suggestion. This sentence has been improved. Line 497-499.